# Monitoring of Respiratory Disease Patterns in a Multimicrobially Infected Pig Population Using Artificial Intelligence and Aggregate Samples

**DOI:** 10.3390/v16101575

**Published:** 2024-10-06

**Authors:** Matthias Eddicks, Franziska Feicht, Jochen Beckjunker, Marika Genzow, Carmen Alonso, Sven Reese, Mathias Ritzmann, Julia Stadler

**Affiliations:** 1Clinic for Swine at the Centre for Clinical Veterinary Medicine, Ludwig-Maximilians-University München, 85764 München, Germany; m.eddicks@lmu.de (M.E.); franziska.feicht@tieraerzte-wonsees.de (F.F.); m.ritzmann@lmu.de (M.R.); 2Boehringer Ingelheim Vetmedica GmbH, Ingelheim, 55216 Ingelheim am Rhein, Germany; jochen.beckjunker@boehringer-ingelheim.com (J.B.); marika.genzow@boehringer-ingelheim.com (M.G.); carmen.alonso@boehringer-ingelheim.com (C.A.); 3Institute for Anatomy, Histology and Embryology, LMU Munich, 80539 Munich, Germany; sven.reese@lmu.de

**Keywords:** surveillance, sample types, novel, diagnostics, respiratory disease, influenza, oral fluids, bioaerosol samples

## Abstract

A 24/7 AI sound-based coughing monitoring system was applied in combination with oral fluids (OFs) and bioaerosol (AS)-based screening for respiratory pathogens in a conventional pig nursery. The objective was to assess the additional value of the AI to identify disease patterns in association with molecular diagnostics to gain information on the etiology of respiratory distress in a multimicrobially infected pig population. Respiratory distress was measured 24/7 by the AI and compared to human observations. Screening for swine influenza A virus (swIAV), porcine reproductive and respiratory disease virus (PRRSV), *Mycoplasma (M.) hyopneumoniae*, *Actinobacillus (A.) pleuropneumoniae*, and porcine circovirus 2 (PCV2) was conducted using qPCR. Except for *M. hyopneumoniae*, all of the investigated pathogens were detected within the study period. High swIAV-RNA loads in OFs and AS were significantly associated with a decrease in respiratory health, expressed by a respiratory health score calculated by the AI The odds of detecting PRRSV or *A. pleuropneumoniae* were significantly higher for OFs compared to AS. qPCR examinations of OFs revealed significantly lower Ct-values for swIAV and *A. pleuropneumoniae* compared to AS. In addition to acting as an early warning system, AI gained respiratory health data combined with laboratory diagnostics, can indicate the etiology of respiratory distress.

## 1. Introduction

Health monitoring is a crucial element in the production of livestock. Next to animal welfare issues and economical losses [1,2,3,4,5], animal health also depicts a major column for the public health sector. A major challenge for pigs’ health is the porcine respiratory disease complex (PRDC) [6]. This disease complex describes a clinical condition associated with treatment-resistant respiratory disease of growing–finishing pigs of a multifactorial etiology, including infectious and noninfectious factors [7]. Porcine circovirus type 2 (PCV2) [8,9], porcine reproductive and respiratory syndrome virus (PRRSV) [6,10], swine influenza A virus (swIAV) [11,12], and *Mycoplasma (M.) hyopneumoniae* [6,13] are considered as the major pathogens involved in the PRDC [6]. Continuous monitoring of animal behavior or evaluation of clinical signs related to pathological changes in the animals’ health is required to reach and maintain a high health status in swine operations. In recent years, several approaches concerning automated analysis of changes in animal behavior or appearance of clinical signs have been published [14,15,16,17,18,19,20,21]. Within this group, wearable devices or sensor technology are assumed to increase animal health and economy in terms of precision livestock farming [20] and thus might even be beneficial concerning human health in terms of the one health strategy of the WHO. In a more concrete way, artificial intelligence (AI) is considered to be helpful to obtain insight into farm infection dynamics [22], detection of clinical signs [14], and decision making [23]. Although automated recognition of animal health-related behavior or clinical signs is thought to be an increasing market [20], the clinical observation of animal health by humans depicts the counterpart and daily work of farmers and veterinarians in the field of livestock farming. Herd inspection often reveals subjectively perceived results unless standardized examinations are carried out. In addition, due to increasing herd sizes, the time for inspection of the animal population or individuals is limited. From this perspective, 24 h automated animal health monitoring seems to be a promising approach that can support the daily work of farmers and covers the time when no human-based inspection is possible. Moreover, it accommodates the possible role of circadian rhythm concerning the inflammatory response on the clinical outcomes of respiratory disease, as reviewed for humans elsewhere [24].

Next to clinical investigations, laboratory diagnostics are needed to gain sufficient information for the implementation of both short-term actions, such as antimicrobial and antiphlogistic treatment, and long-term measures, including vaccinations and management changes. Proper diagnostics and monitoring resemble an important limitation in the control of the porcine respiratory disease complex (PRDC), as the current methods are often labor-intensive, expensive, and based on invasive techniques such as collecting blood samples or sampling the respiratory tract (i.e., nasal swabs, tonsil scratches, tracheobronchial swabs). In the last decade, aggregate samples such as oral fluids (OFs) or environmental samples have gained increasing interest as cost-effective, non-invasive sampling procedures. In particular, OFs are well established for surveillance or monitoring of viral [25,26] or bacterial pathogens [27,28,29], antimicrobial residues, stress, and biomarkers [30,31]. Next to OFs, bioaerosol samples (AS) have been evaluated for surveillance or monitoring of viral pathogens of humans [32] or viral and bacterial pathogens of animals [33,34,35]. Concerning porcine pathogens, PCV2, swIAV [36,37], PRRSV, and *M. hyopneumoniae* [38,39] could be detected in bioaerosols under experimental or field conditions. At least for influenza A, virus bioaerosol sampling revealed similar results compared to oral fluids in a farrow-to-feeder facility [37].

The aim of this longitudinal study was to evaluate the additional value of an AI system for monitoring of respiratory distress in a conventional nursery facility combined with qualitative and quantitative detection of PRDC-associated pathogens in OFs and bioaerosol samples.

## 2. Materials and Methods

### 2.1. Farm Description

This blinded prospective follow-up study was conducted in a conventional one-site piglet-producing farm in Germany with a known history of recurring respiratory distress. The farm was known to be positive for PRRSV, swIAV, *A. pleuropneumoniae*, and *M. hyopneumoniae* based on PCR analysis for diagnostic investigations performed by the herd attending veterinarian. The piglets were vaccinated against PRRS (UNISTRAIN^®^ PRRS, HIPRA, Amer, Spain, i.m.), porcine circovirus diseases (PCVD) (Ingelvac CircoFLEX^®^, Boehringer Ingelheim, Ingelheim am Rhein, Germany), *M. hyopneumoniae*-induced diseases (Hyogen^®^, Ceva, Düsseldorf, Germany) at 21 days of age, and *Lawsonia (L.) intracellularis*-induced diseases (Porcilis^®^ Lawsonia, MSD, Unterschleissheim, Germany) at weaning. Three weeks prior to farrowing, the sows of the corresponding farm were vaccinated with a combined *A. pleuropneumoniae* + *Pasteurella (P.) multocida* and *Streptococcus (S.) suis* autogenous vaccine (SAN VET, Höltinghausen, Germany). In addition, the sows were vaccinated twice a year against swine influenza (Respiporc^®^ FLU3 and Respiporc^®^ FLUpan H1N1, Ceva, Düsseldorf, Germany) and quarterly against PRRS (UNISTRAIN^®^ PRRS, HIPRA, Amer, Spain, i.d.).

The sow farm was producing piglets in a one-week batch farrowing interval. After weaning at 28 days of age, the pigs were transferred to the corresponding nursery facilities. The nursery unit was managed batch-wise, all-in all-out. The nursery unit was designed to house 600 pigs per airspace and consisted of 20 pens with 25–30 pigs per pen on plastic slatted floors. The animals received a commercial diet via a sensor-controlled liquid feeding trough. Water was available ad libitum via bowl- and nipple-drinkers. The ventilation system consisted of a diffused ceiling for supply air and underfloor extraction for exhaust air. The barns were power washed with foam (MS foam A^®^, MS Schippers, Kerken, Germany) and disinfected (MS MegaDes Oxy^®^, MS Schippers, Kerken, Germany) after each batch.

### 2.2. Study Design and Data Collection

For the present study, two consecutive batches (batch 1 (B1), 520 pigs and batch 2 (B2), 519 pigs) in the nursery were enrolled. The day of placement into the nursery unit was considered as study day 0. The piglets in both batches were monitored for 39 days. All procedures were approved by the internal ethical commission of the Veterinary Faculty of the Ludwig-Maximilians-University München (reference 269-24-04-2021).

### 2.3. Automated Cough Monitoring

The commercially available AI device (SoundTalks^®^ NV, Leuven, Belgium) performed continuous and automated measurements of coughs and was able to distinguish between a pigs’ cough and other environmental noises in the barn. It compared the previous day’s coughing level with the current one and calculated a numerical score (respiratory health status; ReHS) with values ranging from 0 (worst case) to 100 (best case). The alerts could be recognized through LED lights on the monitor and through the website. A normal ReHS, which ranges from 60 to 100, was represented by a green signal. Yellow and red alerts indicated a potential and a high risk of a respiratory problem and were equivalent to 40–59 and 0–39 ReHS, respectively. Two SoundTalks^®^ boxes (M1, M2) were installed in the alleyway of the barn. Blinding of the investigator was ensured by setting the LEDs of the monitors to “science mode” (blue LED) over the entire study period (Figure 1). The ReHS was obtained daily (24/7) over the entire study period. The total ReHS displayed the lowest score obtained from one of the two monitors.

### 2.4. Clinical Cough Monitoring

In addition to the continuously automated AI-based cough monitoring, a clinical coughing score (CCS) was determined as published elsewhere [40]. Briefly, the coughing index was calculated as follows:
Coughing index (CI) [%] = Total number of coughing bouts (CC)/[number of examined pigs (n) * total time of observation (min)] * 100

The score was gathered daily within the first five days after weaning. Subsequently, the clinical monitoring was conducted three times a week (Monday, Wednesday, Friday) in each batch until the end of the observational period on day 39. The coughing score was always determined by the same person (the investigator) at the same time during the morning hours.

### 2.5. Laboratory Diagnostic Sampling

Four OFs and two AS were collected (Figure 2) on the same days as the clinical coughing score. For the collection of the oral fluids, a cotton rope (BASKO Aleksander Skoracki, Poznań, Poland; length 100 cm, 3 single cords à Ø1 cm, in total ca. Ø3 cm) was placed for approximately 20 min in each pen next to one of the SoundTalks^®^ monitors. The overall sampling procedure and further processing was performed as published elsewhere [41].

ASs were obtained using two battery-powered AirPrep™ Cub samplers (AirPrep Model ACD210; InnovaPrep, Drexel, MO, USA) according to the sampling scheme of the clinical coughing score. Prior to sampling, a sterile single-use filter (Ø52 mm) made of a dielectric polymer fiber was inserted into each sampler. In the compartment, both samplers were programmed using an intuitive control panel with a flow rate of 200 L/min and a collection time of 60 min. At the end of the 60 min, and with a filtration of 12 m^3^, the sampler stopped automatically. One sampler was placed in the front and one in the back of the study compartment (Figure 1, red diamonds (AS5 and AS6)). To ensure that constant measurement points could be used for both Airprep^™^ cubs, chains were attached to the existing steel beams, and a link was marked. The height of the air sampler was adjusted to the size of the piglets during nursery. In B1, the air inlet opening was located at a height of 77 cm (measured from the floor) from study day 1–26 and at a height of 87 cm from study day 27–39. In B2, the air inlet opening was at a height of 77 cm from study day 1–33 and at a height of 92 cm above the floor from study day 34–39. At the end of the sampling period, the filter was rinsed using the AirPrep Filter and Elution Kit. Briefly, the top of the filter was pressed onto the sample cup, and the elutor adapter was placed on top. Subsequently, the filter was washed out with the elution fluid canister containing TRIS medium using a specially developed procedure via wet foam. This foam broke the electrical tension on the filter surface and disintegrated within a few seconds in the sample cup, resulting in 6 mL of wash solution.

The samples were shipped, cooled and promptly processed upon arrival at the laboratory. All assays were performed in a commercial laboratory using accredited assay methods (Table 1). To extract the genomes, we used KingFisher™ Duo Prime (Thermo Fisher Scientific; Waltham, MA, USA) with an ID Gene™ Mag Universal Extraction Kit (Innovative Diagnostics, Grabels, France). Subsequently, the eluates were frozen and stored in KingFisher™ Elution Strips (Thermo Fisher Scientific; Vantaa, Finland) at −20 °C. After the completion of one run, the selected OFs and AS were analyzed collectively. The “BALF” profile provided by the laboratory was used for this purpose, which included the PCR assays listed in Table 1. The real-time PCR was rated as positive if the Ct-value was ≤40.

### 2.6. Statistical Analysis

The statistical program IBM SPSS Statistics^®^ (version 29.0, IBM^®^ SPSS Inc., Chicago, IL, USA) and Microsoft EXCEL^®^ (version 2019, Microsoft Office, Tulsa, OK, USA) were used for the statistical analysis of the data and to create figures and tables. The significance level was *p* < 0.05. The confidence interval was 0.95.

Continuous data were tested for normal distribution using the Kolmogorov–Smirnov test. Due to non-normal distribution of the data, correlations between CCS, ReHS, the Ct-values of OFs, and the Ct-values of AS were determined using Spearman’s rho. Associations between the non-parametric continuous data Ct-value in dependency of the sample material (OFs and AS) were calculated using the Mann–Whitney U test. Frequency data (detection of pathogens by OFs/AS) were analyzed using cross tables. Fisher’s exact test was used to evaluate the association between the frequency of detection of any pathogen and the sample material. Odds ratios were additionally calculated when appropriate. The level of agreement between OFs and AS for each sampling time was calculated using Cohen’s κ coefficient (κ). Agreement was considered poor if κ ≤ 0.2, fair if 0.21 ≤ κ ≤ 0.4, moderate if 0.41 ≤ κ ≤ 0.6, substantial if 0.61 ≤ κ ≤ 0.8, and good if κ > 0.8 [42].

## 3. Results

### 3.1. Cough Monitoring

The automated AI-based cough monitoring system worked without error, and ReHS could be calculated for every study day. In B1, the first ReHS alert (yellow) was observed on study day 17 only by SoundTalks^®^ monitor 1, located near the entrance, whereas both devices recorded a second alert (yellow) on study day 27 (Figure 3A). In B2, both monitors documented an alert on study day 19; however, monitor 2 already recorded a red alert, whereas monitor 1 showed a yellow alert on day 19 and changed to red on day 20 (Figure 3B).

The total ReHS from both SoundTalks^®^ monitors and the CCS are displayed in Figure 4A for B1 and in Figure 4B for B2. Briefly, in B1, 29/39 days were classified with a good ReHS (green) and 10/39 days with a moderate ReHS (yellow). In B2, 30/39 days were assigned to a good ReHS (green), 2/39 days to a moderate ReHS (yellow), and 7/39 days to a poor ReHS (red).

Twenty human-conducted clinical coughing scores were obtained in each batch. In B1, only mild clinical signs were recorded by the observer, which did not exceed a coughing index of 0.4%. In B2, more severe coughing was obvious, with a maximum coughing index of 1.26% on study day 20.

Spearman’s rho revealed a significant moderate to medium negative correlation between the ReHS and the CCS for B1 (Spearman’s rho −0.478; *p* ≤ 0.001) and B2 (Spearman’s rho: −0.468; *p* ≤ 0.001), indicating that an increase in the CCS was associated with a decrease in the ReHS.

### 3.2. Molecular Biological Examinations

In total, 240 OFs (120 per batch) and 80 AS (40 per batch) were available for PCR analysis. A detailed overview of the detection rate of each pathogen in the two different batches is presented in Table 2.

Briefly, all of the targeted pathogens, with the exception of *M. hyopneumoniae*, could be detected using PCR in both batches. Significant differences concerning the frequency of detection between OFs and AS could be observed for PRRSV and *A. pleuropneumoniae* in both batches (Table 2). In detail, the odds of detecting PRRSV in OFs were 3.72 times higher (95% CI: 1.96–7.08, *p* < 0.001) compared to AS. Moreover, there were 24.89 higher odds (95% CI: 10.17–60.92, *p* < 0.001) of finding *A. pleuropneumoniae* in OFs using PCR compared to AS. In addition, Cohen’s kappa was calculated to evaluate the agreement between OFs and AS to assign the study population as pathogen-positive or negative for each sample day. Only for swIAV a significant agreement (*p* < 0.001) between OFs and AS with a moderate kappa (0.578) could be observed. For all the other pathogens, no significant agreement could be detected. The median Ct-values of the PCR for swIAV-, PRRSV-, PCV2-, and *A. pleuropneumoniae*-positive OFs and AS are shown using boxplots (Figure 5). RNA or DNA loads of *A. pleuropneumoniae* and swIAV were significantly higher in OFs compared to AS (Figure 5), respectively.

The Ct-values of positive OFs and AS over the entire study period are shown in Figure 6.

To evaluate potential relationships between the extent of coughing (expressed by ReHS or CCS) and the viral or bacterial loads (expressed by Ct-values) in OFs or AS, bivariate correlations (Spearman’s rho) were calculated. A significant correlation was observed between the Ct-values of the swIAV PCR results and the ReHS (Spearman’s rho: AS: 0.630, *p* < 0.001; OFs: 0.693, *p* < 0.001) and CCS (Spearman’s rho: AS: −0.542, *p* < 0.001, OFs: −0.535, *p* < 0.001) for the entire study period. Figure 7 depict the Ct-values of swIAV for OFs and AS in correlation with the ReHS (Figure 7A) or CCS (Figure 7B), including the regression equation for swIAV Ct-values depending on the used sampling material.

## 4. Discussion

Early detection of respiratory distress is essential to preserve animal health and to combat economic losses due to infections with respiratory pathogens. Within the present study, we combined a sound-based AI respiratory health monitoring system with a molecular biological laboratory screening based on OFs and AS for the detection of pathogens associated with PRDC. Within this scope, we also compared OFs and AS as matrices for the detection of nucleic acids of PRRSV, PCV2, APP, and swIAV. As observed by others [43], the AI-based monitoring system proved to be technically reliable, and respiratory health data could be obtained for each study day. However, limitations concerning the use of this monitoring might include a lack of internet within the range of the stables, which should be checked prior to the installation of such a monitoring system. The comparison between the 24/7 AI and manual coughing monitoring correlated on a moderate to medium level, as also reported by others [16]. In principle, this correlation was expected, as coughing is needed to register a corresponding response by the AI monitoring system. However, the ReHS gave more detailed information, e.g., in the first batch when coughing was present, particularly in the front part of nursery unit. Thus, next to the recognition of clinical signs, AI-based long-term monitoring on farms might also demonstrate localized patterns of disease that indicate issues concerning the farm environment or internal biosecurity. The advantage of continuous AI-based monitoring becomes clearly recognizable as different times of activity of pigs over a daily period can be addressed, whereas a clinical examination is limited in time and only covers the time of the physical presence in the corresponding pig population. This might explain the only moderate correlation between ReHS and CCS in the present examination. The circadian rhythm of the pigs in their surrounding environment influences the times of activity of the pigs [44] and thus the appearance of clinical signs due to the circadian oscillation of the immune system. In terms of respiratory diseases in humans, the severity of clinical signs also shows circadian variability across a 24 h cycle. Increased inflammation and disease severity at night is described for obstructive airway diseases and allergic rhinitis, with the consequence of greater effects of exposure to inflammatory insults at night [24]. Based on a previous investigation, the optimal number of devices for the 600-head barn was determined to be two [45]. However, to ensure optimal sound coverage and to account for variations in infection dynamics, the number of required monitors and the placement of the devices must be thoroughly considered.

The diagnostic screening using PCR revealed the presence of multiple respiratory pathogens in the nursery of the study farm. Thus, the principal preconditions for PRDC were present in the study population. Nevertheless, in order to definitively determine whether a certain combination of pathogens contributed to the clinical signs, pathomorphological examinations would have been required.

Interestingly, a significant correlation was observed between decreasing Ct-values of the swIAV PCR and increasing respiratory distress measured by the AI or the investigator. Thus, an increase in the swIAV viral load in OFs or AS coincided with the extent of the clinical signs in terms of coughing, expressed by reduced ReHS or increased CCS, respectively. Comparable observations were made by Neira et al. [46] who reported a correlation between the quantitative detection of swIAV in oral fluids and the coughing score. Although AI monitoring systems are not able to distinguish between different pathogens, previous initial investigations have shown a bi-modal respiratory distress pattern for swIAV infections in three commercial wean-to-finish facilities [47]. The link between swIAV viral load in OFs or AS and the extent of coughing in our observation also provides evidence to suggest that swIAV might have been the driving force for respiratory distress in this nursery unit. Moreover, based on the comparable pattern of swIAV RNA detection in both batches, an endemic swIAV infection can be assumed. As already postulated by Prost et al. [37] and Anderson et al. [36], no significant differences concerning the qualitative detection of swIAV using PCR were evident between OFs and bioaerosol samples. However, in contrast to Prost et al. [37], swIAV RNA loads were significantly higher in OFs compared to bioaerosol samples.

The detection of PRRSV might either be a result of the MLV vaccination of the piglets or of an infection with a low virulent strain, as no significant correlation between clinical observations and PRRSV detection was observed in the present study. However, further diagnostics to discriminate between field or vaccine strains were not successful due to the low viral loads in both sampling materials. Although airborne transmission of PRRSV over more than four kilometers was reported [38], the detection rate of viral nucleic acids in bioaerosol samples compared to OFs in our study was rather low. This observation indicates less suitability of bioaerosols for monitoring or surveillance purposes of PRRSV compared to OFs whose sensitivity and suitability to monitor PRRSV were already shown elsewhere [48,49,50]. Still, these results should be interpreted with caution, because airborne transmission of PRRSV might be strain-dependent, as reported by others [39], and the attenuation of the vaccine strain that was putatively detected here might bias our findings.

Significant advantages of OFs over AS could also be observed for the detection of *A. pleuropneumoniae*. The high sensitivity of OFs concerning the detection of *A. pleuropneumoniae* using PCR was already shown in a previous study under field conditions [27]. The sporadic detection of *A. pleuropneumoniae* DNA in the bioaerosols is in line with the limited capability of *A. pleuropneumoniae* aerosol transmission reported elsewhere [51] and the higher within-pen transmission rate compared to the transmission between different pens [52]. However, the sporadic detection of APP in the bioaerosols indicates the possibility of airborne infections with *A. pleuropneumoniae* within a pig population, as shown by others [51,52]. Moreover, differences in the ventilation rate, air humidity, particle size, and the overall bacterial load in the pig population might influence the detection of bacteria in bioaerosol samples. Particularly, the underfloor extraction for the exhaust air of the stable in the present study might contribute to a reduced amount of bacteria in the air, which could also apply to a certain extent to *A. pleuropneumoniae*. No association between clinical disease patterns and qualitative or quantitative detection of *A. pleuropneumoniae* were obvious in the present study. Possibly, a low virulent strain was circulating in the herd, or the pigs were subclinically infected. However, the applied PCR did not allow for conclusions on the virulence of the circulating strain/s. Moreover, maternally derived immunity, which can last up to 12 weeks of age, might also have prevent clinical disease in our study herd [53].

Concerning PCV2, Harms et al. [8] demonstrated the relevance of the combination of PCV2 and swIAV in a PRDC case series. However, in the present study, the qualitative and quantitative detection rate of PCV2 was low in both sample types. Although Nielsen et al. [54] postulated that PCV2 detection in OFs are not necessarily correlated with clinical signs, PCV2 does not seem to play a major role concerning the clinical outcome in the present nursery, as evident from Figure 5. Moreover, the pigs in the present examination were vaccinated against PCV2. The positive effects of PCV2 vaccination concerning PRDC or PCV2 systemic disease-affected herds [55,56] on clinical outcomes and co-infections are widely known.

Although the farm was categorized as *M. hyopneumoniae* positive and samples were obtained on numerous occasions, *M. hyopneumoniae* DNA was not detected in any of the collected samples. Based on the literature, the detection of *M. hyopneumoniae* DNA from OFs does not appear to have the same sensitivity as the detection of the pathogens mentioned above. This was shown both in studies under experimental conditions and under field conditions. In their challenge test, Pieters et al. [57] examined a wide variety of materials for the detection of *M. hyopneumoniae* DNA. Laryngeal swabs performed best, while OFs were described as occasionally positive, which was already indicated in a previous study [58]. Under field conditions, Hernandez-Garcia et al. [25] recommended several examination time points to compensate for the disadvantages of inconsistent detection. Studies under field conditions at least indicated that OFs at the end of fattening, in association with clinical signs (coughing), appear suitable for the detection of *M. hyopneumoniae* DNA [29]. Furthermore, in the aforementioned study, there was a significant positive correlation between the amount of DNA in the OFs and the extent of lung lesions that could be attributed to enzootic pneumonia. Concerning AS, studies have shown the possibility of detecting *M. hyopneumoniae* DNA in AS under field conditions [38,59]. However, as already mentioned concerning APP, ventilation rate, air humidity, particle size, and bacterial load might influence the detection of bacteria in bioaerosol samples. Particularly, the underfloor extraction for the exhaust air in the present study might contribute to a reduced amount of bacteria in the air. Likely, a low infection pressure in the nursery, in combination with a medium sensitivity of the used materials under the study conditions, resulted in negative test results. Moreover, a late onset of *M. hyopneumoniae* infection and disease in downstream production might also be of relevance for our findings; however, as no samples were obtained from finishers, we cannot draw a conclusion.

Although aggregate sampling is an accepted tool for monitoring and surveillance of respiratory pathogens, it does not necessarily allow conclusions on the clinical relevance of the detected pathogens. Here, the additional application of an AI respiratory health monitoring system delivers further benefits by indicating the primary etiology of respiratory distress and thus enabling timely diagnostics and treatment. However, further development of the algorithms is needed to allocate coughing patterns to other respiratory pathogens.

## 5. Conclusions

The AI-based coughing monitoring (SoundTalks^®^) used in the present examination reliably delivered detailed data on the pigs’ respiratory health and showed localized disease pattern in the nursery. The correlation between the extent of the respiratory distress, expressed by ReHS, and the quantity of swIAV RNA in OFs and AS allowed for speculations on the primary etiology of the respiratory distress. Under the present study conditions, bioaerosol sampling was less sensitive to detecting *A. pleuropneumoniae* and PRRSV using PCR compared to oral fluids. Moreover, *A. pleuropneumoniae* bacterial and swIAV viral loads were significantly higher in oral fluids compared to AS. The application of a long-term AI-based coughing monitoring system in combination with laboratory examinations can be a useful tool to identify relevant pathogens and recurring disease patterns on farms and to recognize respiratory distress under field conditions early.

## Figures and Tables

**Figure 1 viruses-16-01575-f001:**
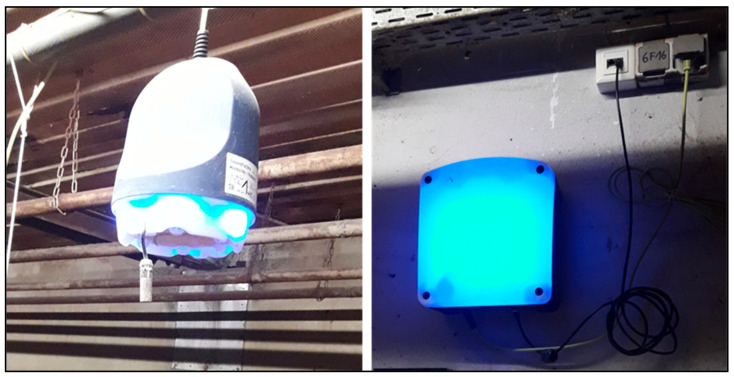
Left picture: SoundTalks^®^ monitor in science mode (blue led). Right picture: Plugged gateway in science mode (blue led).

**Figure 2 viruses-16-01575-f002:**
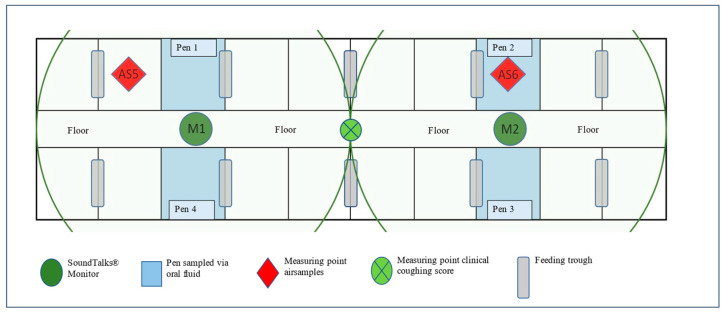
Outline of the study compartment and position of oral fluids and air sample collection points, clinical coughing score assessment, and SoundTalks^®^ monitor (M1 and M2). The radius lines correspond to the coverage range (10 m) of the respective SoundTalks^®^ monitor (M1 and M2).

**Figure 3 viruses-16-01575-f003:**
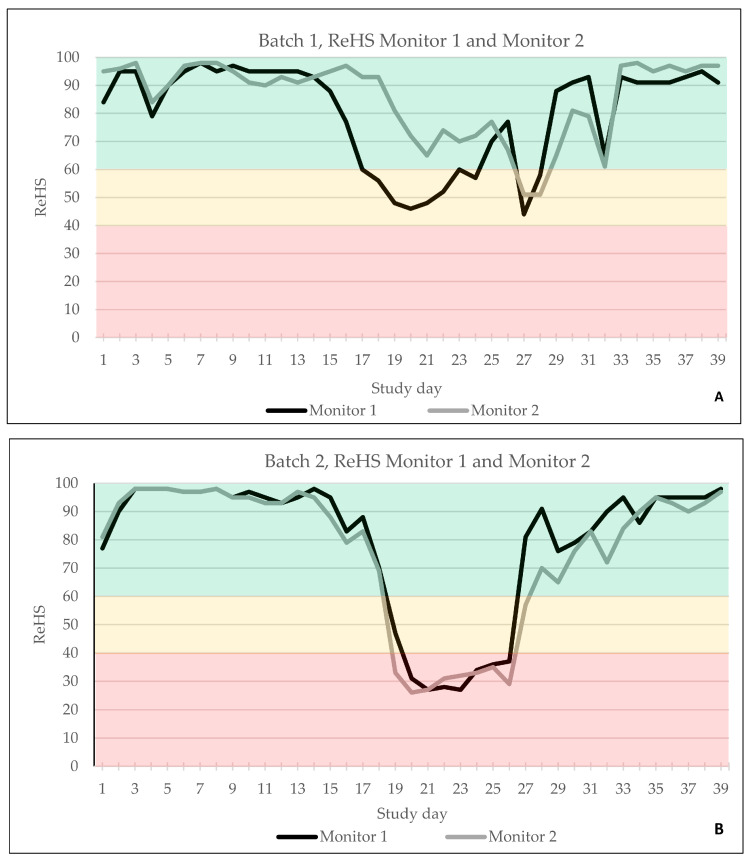
ReHS of batch 1 (**A**) and batch 2 (**B**) obtained by monitor 1 and monitor 2 over the entire study period. Green indicated a normal ReHS, which ranges from 60 to 100. Yellow indicated a potential risk of a respiratory problem, equivalent to 40–59 ReHS and red a high risk of a respiratory problem, equivalent to 0–39 ReHS.

**Figure 4 viruses-16-01575-f004:**
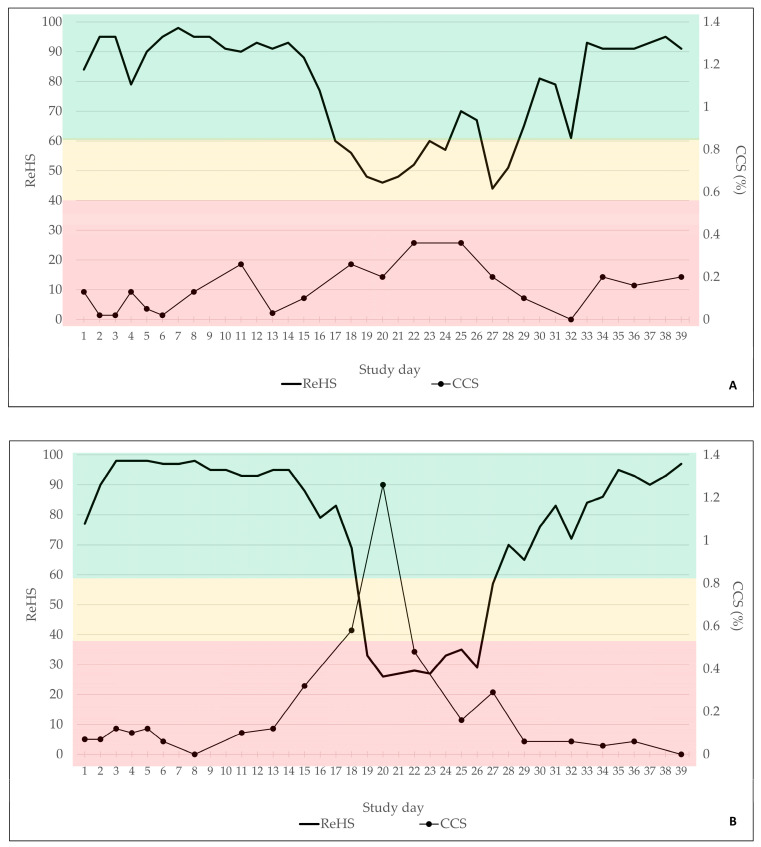
Total ReHS and CCS of batch 1 (**A**) and batch 2 (**B**) for the observational period of 39 days. Green indicated a normal ReHS, which ranges from 60 to 100. Yellow indicated a potential risk of a respiratory problem, equivalent to 40–59 ReHS and red a high risk of a respiratory problem, equivalent to 0–39 ReHS.

**Figure 5 viruses-16-01575-f005:**
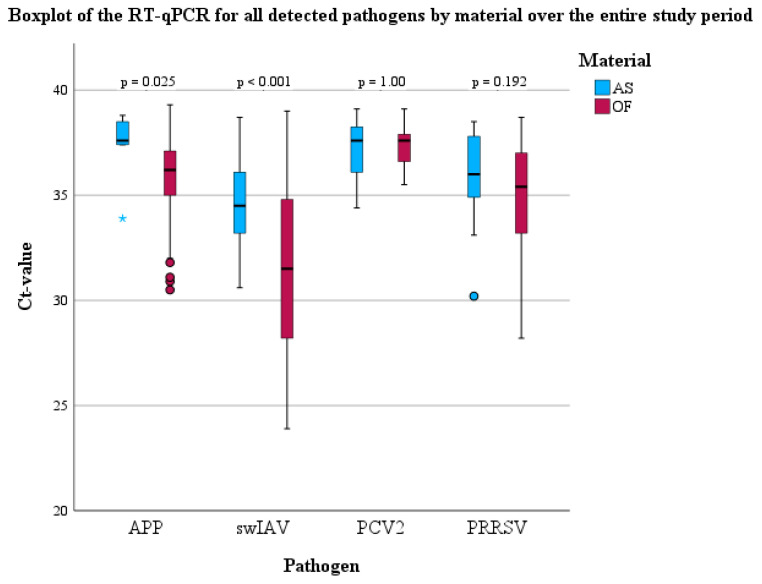
Boxplots (median, lower, and upper quartile, minimum and maximum) of the PCR results for all detected pathogens in OFs and AS over the entire study period.

**Figure 6 viruses-16-01575-f006:**
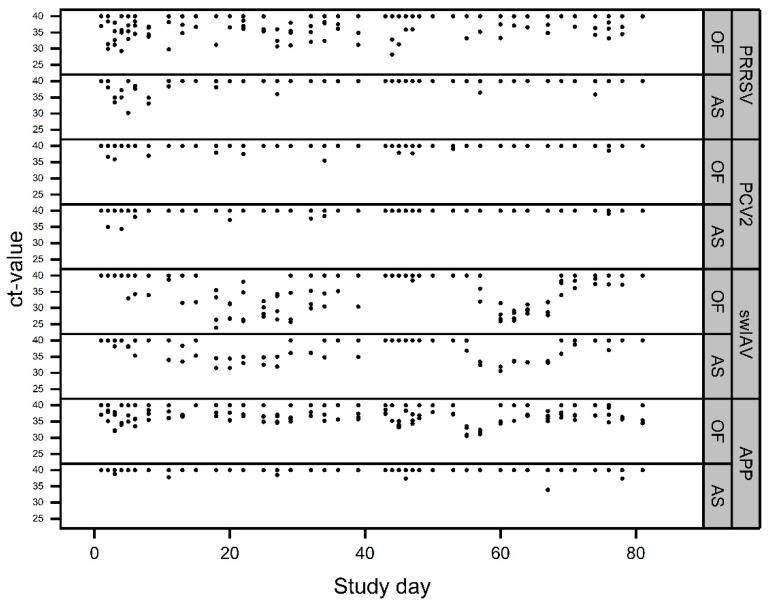
Detection patterns of the different pathogens over the entire study period (batch 1: study days 0–39; batch 2: study days 40–80) in OFs and AS. Ct-values ≥ 40 represent negative results. Overlaying results are possible.

**Figure 7 viruses-16-01575-f007:**
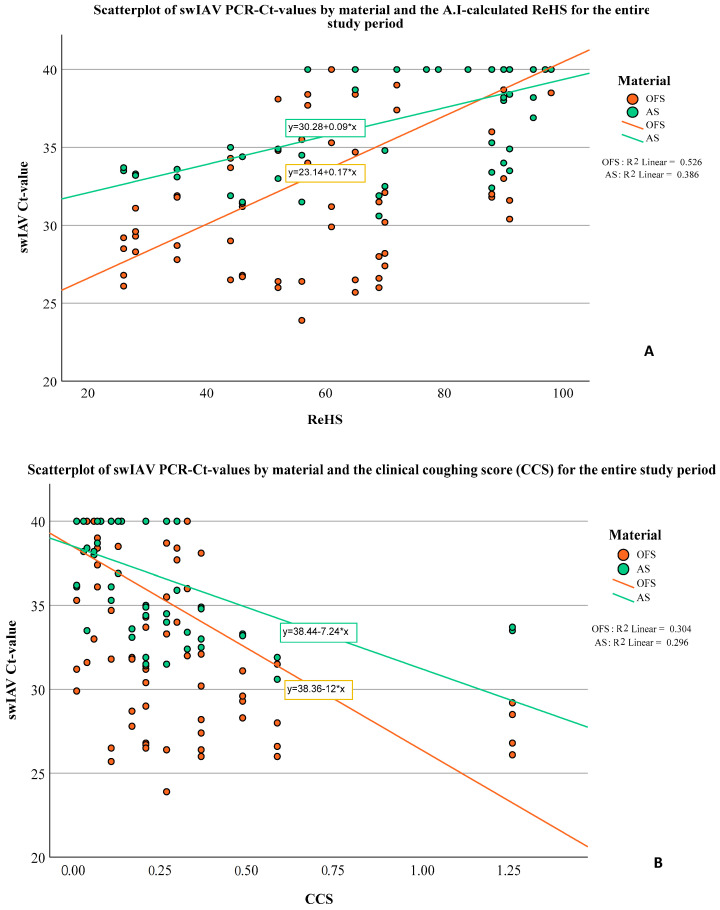
Scatterplot of swIAV PCR Ct-values in correlation with the ReHS (**A**) or CCS (**B**) for the entire study period for AS and OFs and the corresponding regression equation.

**Table 1 viruses-16-01575-t001:** Name and manufacturer of the assays used in the present study.

Pathogen	Name of the Used Assay	Manufacturer
PRRSV	Virotype PRRSV NA/EU	INDICAL BIOSCIENCE GmbH; Leipzig, Germany
PCV2	Virotype PCV2/PCV 3	INDICAL BIOSCIENCE GmbH; Leipzig, Germany
swIAV	Virotype Influenza A RT-PCR	INDICAL BIOSCIENCE GmbH; Leipzig, Germany
*M. hyopneumoniae*	EXOone *Mycoplasma hyopneumoniae*	exopol; San Mateo de Gállego, Zaragoza, Spain
*A. pleuropneumoniae*	EXOone *Actinobacillus pleuropneumoniae*	exopol; San Mateo de Gállego, Zaragoza, Spain

**Table 2 viruses-16-01575-t002:** Percentage (%) and frequency of detection (n) of specific pathogens using PCR in OFs and AS of batch 1 and batch 2 and for the entire study period, as well as *p*-values in the case of significant differences in the frequency of detection.

	B1			B2			Entire Study Period		
	OFs (n = 64)	AS (n = 32)	*p*-Value	OFs (n = 60)	AS (n = 30)	*p*-Value	OFs (n = 124)	AS (n = 62)	*p*-Value
PRRSV	65.0% (52/80)	32.5% (13/40)	<0.001	27.5% (22/80)	5.0% (2/40)	0.003	46.3% (74/160)	18.8% (15/80)	<0.001
PCV2	7.5% (6/80)	15.0% (6/40)	0.211	5.0% (4/80)	2.5% (1/40)	0.664	6.3% (10/160)	8.8% (7/80)	0.594
swIAV	45.0% (36/80)	57.5% (23/40)	0.246	36.3% (29/80)	35.0% (14/40)	1.00	40.6% (65/160)	46.3% (37/80)	0.410
APP ^1^	63.7% (51/80)	7.5% (3/40)	<0.001	70.0% (56/80)	7.5% (3/40)	<0.001	66.9% (107/160)	7.5% (6/80)	<0.001
M. hyo ^2^	0.0% (0/64)	0.0% (0/32)	-	0.0% (0/60)	0.0% (0/30)	-	0.0% (0/124)	0.0% (0/62)	n.d. *

* n.d.: not detected, ^1^
*A. pleuropneumoniae*, ^2^
*M. hyopneumoniae*.

## Data Availability

The dataset is available from the corresponding author upon request.

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
