# Peer review of "Monitoring of Respiratory Disease Patterns in a Multimicrobially Infected Pig Population Using Artificial Intelligence and Aggregate Samples"

_viruses, 2024, doi:10.3390/v16101575_

Round 1

Reviewer 1 Report

Comments and Suggestions for Authors

The present paper investigates the suitability of a 24/7 A.I. sound based coughing monitoring 12 system for early detection of PRDC in swine farms. The study employs clinical observations by a veterinarian and diagnostic assays (RT-PCR) to investigate the agreement (agreement study) between the A.I. monitoring system and “traditional” PRDC monitoring methods. Overall, the article is well written and clear, and contributes to the popularization of novel tools and technologies for the monitoring of animal diseases. Minor text editing is required to follow the journal’s instructions. It is also recommended to address the following comments:

1.    Abstract: The abstract needs to be revised. The objective, methodology, results and conclusions are not very clear in the abstract prior reding the main text.

2.    Discussion: The title of the article is “Artificial intelligence and innovative surveillance methods - an additional value in livestock farming?”. Add a paragraph at the end of the discussion that answers the question posed in the title. Include the added value of the technology (it is described throughout the discussion, but it should also be summarized at the end) and use this specific tool as an example to make some general remarks/recommendations about the use of AI and innovative surveillance methods in livestock farming.

3.    L59: (….long-term measures as vaccinations…) probably you are missing some words

4.    L173: Please revise. It is not clear what “Error! Reference source not found” means.

5.    L204 – L237: Follow the text formatting instructions of the journal. There are additional blank paragraphs.

6.    L231: Revise the text.

7.    L244-248: Follow the text formatting instructions of the journal. There are additional blank paragraphs.

8.    L240 & L251: Use capital T when you refer to Table 2 in the text.

9.    L259, L260, L266, L368: Use capitals when you refer to tables or figures in the text.

10. L276-285: The text and figures do not add too much value in the manuscript apart from visualizing the results described in L271-276, especially given that the linear models do not fit well the data. You can consider removing them (optional).

11. References: Use the appropriate text format and citation style according to the journal’s instructions.

Comments on the Quality of English Language

Minor editing of English language required at specific lines (check Comments and Suggestions for Authors).

Author Response

The present paper investigates the suitability of a 24/7 A.I. sound based coughing monitoring system for early detection of PRDC in swine farms. The study employs clinical observations by a veterinarian and diagnostic assays (RT-PCR) to investigate the agreement (agreement study) between the A.I. monitoring system and “traditional” PRDC monitoring methods. Overall, the article is well written and clear, and contributes to the popularization of novel tools and technologies for the monitoring of animal diseases. Minor text editing is required to follow the journal’s instructions. It is also recommended to address the following comments:

We are very grateful for the overall positive evaluation of our manuscript

  1. Abstract: The abstract needs to be revised. The objective, methodology, results and conclusions are not very clear in the abstract prior reading the main text.

We thank the reviewer for the suggestion. We have revised the abstract in order to clarify the objective, methodology, results and conclusions.

  1. Discussion: The title of the article is “Artificial intelligence and innovative surveillance methods - an additional value in livestock farming?”. Add a paragraph at the end of the discussion that answers the question posed in the title. Include the added value of the technology (it is described throughout the discussion, but it should also be summarized at the end) and use this specific tool as an example to make some general remarks/recommendations about the use of AI and innovative surveillance methods in livestock farming.

As requested by reviewer 3 we have changed the title to “Monitoring of respiratory disease patterns in a multimicrobial infected pig population through artificial intelligence and aggregate samples”. We totally agree with the reviewer that a summary at the end of the discussion is of great value for the manuscript. Therefore, we have added general remarks/recommendations about the use of AI and innovative surveillance methods in livestock farming at the end of the discussion.

  1. L59: (….long-term measures as vaccinations…) probably you are missing some words

We have amended the sentence accordingly: Next to clinical investigations, laboratory diagnostics are needed to gain sufficient information for the implementation of both short-term actions such as antimicrobial and antiphlogistic treatment, and long-term measures including vaccinations and management changes.

  1. L173: Please revise. It is not clear what “Error! Reference source not found” means.

Please excuse the error. There was an unintentional link to table 1 that was not visible in my submitted version of the manuscript

  1. L204 – L237: Follow the text formatting instructions of the journal. There are additional blank paragraphs.

We have now added the figures according to formatting instructions.

  1. L231: Revise the text.

Thank you for your advice. We did accordingly.

  1. L244-248: Follow the text formatting instructions of the journal. There are additional blank paragraphs.

We have now added the figures according to formatting instructions.

  1. L240 & L251: Use capital T when you refer to Table 2 in the text.

We have used capital letters for tables in the text

  1. L259, L260, L266, L368: Use capitals when you refer to tables or figures in the text.

We have used capital letters for tables and figures in the text

  1. L276-285: The text and figures do not add too much value in the manuscript apart from visualizing the results described in L271-276, especially given that the linear models do not fit well the data. You can consider removing them (optional).

Dear Reviewer, we would be happy to leave them in the manuscript, as it delivers more detailed information on the viral load of single OFs and AS in correlation to the ReHS and CCS.

  1. References: Use the appropriate text format and citation style according to the journal’s instructions.

We have now incorporated the citation style of the Journal

Reviewer 2 Report

Comments and Suggestions for Authors

The main aim of the paper titled Artificial intelligence and innovative surveillance methods - an additional value in livestock farming? was to present data obtanied from a AI based coughing monitoring system implemented at a commercial swine farm and combine them with laboratory reports delineating detection of specific pathogens responsible for porcine respiratory diseases. The manuscript presented by the authors is clear, well-written and very interesing. The experimental desing is appropriate. The conclusions are presented well and align with the results obtained during the investigation.

The text requires minor revisions:

L18: odds, not Odds

L19-20: pleuropneumoniae, not pleuropneumonia

L25: influenza, not Influenza

L64-67: Indeed, oral fluid samples have gained increased interest. Moreover, the use of the matrix in swine medicine goes beyond the surveillance of bacterial or viral pathogens. Please provide the readers with such information (e.g. acute phase proteins, antibiotics).

L71-72: Please rewrite the sentence to avoid the repetition (comparable, compared).

L82-90: The piglets were vaccinated against diseases (e.g. porcine circoviral disease), not against pathogens (PCV 2). Please rewrite this part.

L84: Please replace the second bracket (between Hyogen and Ceva) with a comma.

L85, L90: The piece of information regarding commercially available vaccines is redundant.

L88: Please provide the information regarding the manufacturer of the vaccine.

L94: Please remove the spaces in all- in all- out.

L107: Is it possible to provide the readers with a photograph of the devices?

L136: Please use the original city name, as stated by the producer on the package.

L146: Please complete the company location and be consistent in that regard (e.g. L168).

L173: Please correct the reference error.

L180: Please replace the semicolon (Illinois; USA).

L188: Please correct the name of the test.

L206: Can you expand the abbreviation? The comment applies to all the other figures too. I believe it will improve readability of the text. 

L232-233: Please be consistent with the notation: -0.478 and 0.001, not -.478 and 0.001

L244: The space in the line with APP is missing (< 0.001, not<0.001).

L254: This is kappa, not Kappa.

L258: Two hyphens are missing.

L319-371: Can you divide the paragraph into smaller ones just to improve the readability of the text?

L361-363: Particularly the underfloor extraction for the exhaust air of the stable in the present study might contribute to a reduced the number of bacteria in the air, which could also in a certain extend apply to A. pleuropneumoniae.

L401: Plase see the journal reference formatting guide and double-check the list.

Author Response

Reviewer 2:

The main aim of the paper titled Artificial intelligence and innovative surveillance methods - an additional value in livestock farming? was to present data obtanied from a AI based coughing monitoring system implemented at a commercial swine farm and combine them with laboratory reports delineating detection of specific pathogens responsible for porcine respiratory diseases. The manuscript presented by the authors is clear, well-written and very interesing. The experimental desing is appropriate. The conclusions are presented well and align with the results obtained during the investigation.

We are very grateful for the overall positive evaluation of our manuscript

The text requires minor revisions:

L18: odds, not Odds

We have corrected the spelling mistake accordingly.

L19-20: pleuropneumoniaenot pleuropneumonia

We have corrected the spelling mistake accordingly.

L25: influenza, not Influenza

We have corrected the spelling mistake accordingly.

L64-67: Indeed, oral fluid samples have gained increased interest. Moreover, the use of the matrix in swine medicine goes beyond the surveillance of bacterial or viral pathogens. Please provide the readers with such information (e.g. acute phase proteins, antibiotics).

We would like to thank the author for this suggestion. We have added the use of oral fluids for monitoring of antimicrobial residues and their implementation in detection of stress and biomarkers.

L71-72: Please rewrite the sentence to avoid the repetition (comparable, compared).

We have amended the sentence.

L82-90: The piglets were vaccinated against diseases (e.g. porcine circoviral disease), not against pathogens (PCV 2). Please rewrite this part.

We have rewritten the section about vaccination accordingly.

L84: Please replace the second bracket (between Hyogen and Ceva) with a comma.

We have replaced the bracket with a comma.

L85, L90: The piece of information regarding commercially available vaccines is redundant.

We have deleted the wording commercially available.

L88: Please provide the information regarding the manufacturer of the vaccine.

We have added the manufacturer of the vaccine.

L94: Please remove the spaces in all- in all- out.

We have removed the spaces.

L107: Is it possible to provide the readers with a photograph of the devices?

Thank for your suggestion. We added photos that we had from our study.

L136: Please use the original city name, as stated by the producer on the package.

We have changed the name of the city from Posen to PoznaÅ„. 

L146: Please complete the company location and be consistent in that regard (e.g. L168).

We have added the city where the company is located and deleted the federal state in order to be consistent throughout the manuscript.

L173: Please correct the reference error.

Please excuse the linking error to table 1.

L180: Please replace the semicolon (Illinois; USA).

We have replaced the semicolon with a comma.

L188: Please correct the name of the test.

We have corrected the name accordingly.

L206: Can you expand the abbreviation? The comment applies to all the other figures too. I believe it will improve readability of the text. 

We totally agree with the reviewer and have expanded the abbreviation to batch 1 and batch 2 in all figures.

L232-233: Please be consistent with the notation: -0.478 and 0.001, not -.478 and 0.001

We have changed it accordingly.

L244: The space in the line with APP is missing (< 0.001, not<0.001).

We have inserted a line.

L254: This is kappa, not Kappa.

We have corrected the spelling mistake accordingly.

L258: Two hyphens are missing.

We have added the two hyphens.

L319-371: Can you divide the paragraph into smaller ones just to improve the readability of the text?

Thank you so much for your suggestion. To improve the readability of this part of the discussion we have inserted paragraphs.

L361-363: Particularly the underfloor extraction for the exhaust air of the stable in the present study might contribute to a reduced the number of bacteria in the air, which could also in a certain extend apply to A. pleuropneumoniae.

Thank you so much for noticing this error, we have corrected it accordingly.

L401: Please see the journal reference formatting guide and double-check the list.

We have now formatted the references according to the style of the journal.

Reviewer 3 Report

Comments and Suggestions for Authors

Dear authors!

In your manuscript you describe a AI technology measuring coughing events in pigs comparison with conventional surveillance techniques (OF and AS).

I gave some feedback directly in the PDF version of the manuscript.

Some points I would like to mention separately:

1. The title is not reflecting the content of the manuscript adequately. I would change it, so the reader can imagine the content just from reading the title.

2. PRRSV diagnostics was done using PCR. However, I would like to ask you again, if you may try to get at least any sequence data on the strain/strains circulating in the study farm. Maybe it is worth trying to sequence ORF7 and/or ORF5 to have at least any information available. At the moment discussion about PRRSV is based on MLV strains only.

3. A major point, that has to be improved in a revised version is the fact, that absolutely no discussion on the negative results about Mycoplasma hyopneumoniae can be found. Why was it just ignored? Please discuss the M hyo results.

Comments on the Quality of English Language

In some parts the order of words in sentences is not appropriate. However, I think it is not my duty as a reviewer to correct English style. Please try to contact a native speaker. Maybe Oliver Duran from BI would be available proofreading the manuscript.

Author Response

  1. The title is not reflecting the content of the manuscript adequately. I would change it, so the reader can imagine the content just from reading the title.

We would like to thank the reviewer for the suggestion. We have changed the title to: “Monitoring of respiratory disease patterns in a multimicrobial infected pig population through artificial intelligence and aggregate samples”.

  1. PRRSV diagnostics was done using PCR. However, I would like to ask you again, if you may try to get at least any sequence data on the strain/strains circulating in the study farm. Maybe it is worth trying to sequence ORF7 and/or ORF5 to have at least any information available. At the moment discussion about PRRSV is based on MLV strains only.

We thank the reviewer for the suggestion. Due to the comment of the reviewer we have now initiated discrimination between field and vaccine strain via PCR. However, the PCR was not successful due to low viral loads that were now even lower due to the storage of the samples.

  1. A major point, that has to be improved in a revised version is the fact, that absolutely no discussion on the negative results about Mycoplasma hyopneumoniae can be found. Why was it just ignored? Please discuss the M hyo results.

We totally agree with the reviewer that a discussion on Mycoplasma hyopneumoniae is needed. We sincerely apologize for the omission. We have added a discussion about the negative results.

Why not discussing, that APP is not clinically affecting pigs in the study farm? So far, swIAV is obviously endemic in this farm and may be the primary cause of coughing there. Otherwise one would expect higher rates of losses due to APP.....Think about my comment and include it in the discussion if it finds consent...

We have expanded the discussion on A. pleuropneumoniae.  

Additional comments in the pdf:

We have changed the term Mesomycoplasma to Mycoplasma

We have consistently used the term bioaerosol samples

We have replaced the parenthesis with a comma

We have added more details on the Cleaning and Disinfection protocol (Name of disinfectant, name of foam)

Please excuse the spelling error. We now have consistently used the abbreviation OFs

Please excuse the error regarding the link to table 1

For the presents study we only considered PCV-2 as a relevant pathogen for respiratory diseases

We have changed the sentence to: „Odds ratio additionally were calculated when appropriate

We have inserted a continuous line in table 2

Please excuse the spelling error, we have corrected the word extent

The dot was inserted (A.I.)

We have deleted the word “the“

We have changed PMWS to PCV2-systemic disease

Round 2

Reviewer 3 Report

Comments and Suggestions for Authors

Dear Authors!

Thanks for taking my recommendations, corrections and suggestions into account. I read your revised mansucript and think you were able to improve it mostly. However, I had some comments I think that still can improve your manuscript even more.

The comments can be found directly in the last version that was provided to me.

Author Response

We are very grateful for the comments of the reviewer as they helped to improve the content of the manuscript. We have implemented all suggestions and comments of the reviewer.

We have correct the error occuring in the link to table 1.

We have changed the wording from PCV-2 to PCV2 throughout the manuscript

We have formatted the refernces according to the style of the journal

We have expanded the discussion on the additional potential application of A.I.